# The Use of Winery by-Products to Enhance the Functional Aspects of the Fresh Ovine “Primosale” Cheese

**DOI:** 10.3390/foods10020461

**Published:** 2021-02-20

**Authors:** Raimondo Gaglio, Pietro Barbaccia, Marcella Barbera, Ignazio Restivo, Alessandro Attanzio, Giuseppe Maniaci, Antonino Di Grigoli, Nicola Francesca, Luisa Tesoriere, Adriana Bonanno, Giancarlo Moschetti, Luca Settanni

**Affiliations:** 1Dipartimento Scienze Agrarie, Alimentari e Forestali, Ed. 5, Università degli Studi di Palermo, Viale delle Scienze, 90128 Palermo, Italy; pietro.barbaccia@unipa.it (P.B.); marcella.barbera@unipa.it (M.B.); giuseppe.maniaci@unipa.it (G.M.); antonino.digrigoli@unipa.it (A.D.G.); nicola.francesca@unipa.it (N.F.); adriana.bonanno@unipa.it (A.B.); giancarlo.moschetti@unipa.it (G.M.); luca.settanni@unipa.it (L.S.); 2Dipartimento di Scienze e Tecnologie Biologiche, Chimiche e Farmaceutiche, Università degli Studi di Palermo, Via Archirafi 34, 90123 Palermo, Italy; ignazio.restivo@unipa.it (I.R.); alessandro.attanzio@unipa.it (A.A.); luisa.tesoriere@unipa.it (L.T.)

**Keywords:** functional ovine cheese, grape pomace powder, *Lactococcus lactis*, physicochemical properties, polyphenols, volatile organic compounds, antioxidant properties

## Abstract

Fresh ovine “primosale” cheese was processed with the addition of grape pomace powder (GPP). Cheese making was performed using pasteurized ewes’ milk and four selected *Lactococcus lactis* strains (Mise36, Mise94, Mise169 and Mise190) inoculated individually. For each strain the control cheese (CCP) was not added with GPP, while the experimental cheese (ECP) was enriched with 1% (*w*/*w*) GPP. GPP did not influence the starter development that reached levels of 10^9^ CFU/g in all final cheeses. The comparison of the bacterial isolates by randomly amplified polymorphic DNA (RAPD)-PCR showed the dominance of the added strains over indigenous milk bacteria resistant to pasteurization. GPP addition reduced fat content and determined an increase of protein and of secondary lipid oxidation. Sensory tests indicated that cheeses CCP94 and ECP94, produced with the strain Mise94, reached the best appreciation scores. Following in vitro simulated human digestion, bioaccessible fraction of ECP94 showed antioxidant capacity, evaluated as radical scavenging activity and inhibition of membrane lipid oxidation, significantly higher than that from CCP94, with promising increase in functional properties. Thus, the main hypothesis was accepted since the functional aspects of the final cheeses improved, confirming that GPP is relevant for sustainable nutrition by using winemaking by-products.

## 1. Introduction

The production of wine generates a large amount of by-products, known as grape pomace, that are composed by a mix of grape skins and seeds [1]. Grape pomace constitutes a relevant environmental issue related to the production of wine [2] even though it represents a consistent source of functional compounds such as polyphenols and dietary fiber [3]. These compounds can exert a positive impact on human health, especially for the prevention of diseases associated with oxidative stress such as cancer, stroke and coronary heart disease [4]. Others main constituents of grape pomace are colorants, minerals and organic acids [5,6]. Moreover, considering that grape pomace is classified as a Generally Recognized As Safe (GRAS) matrix by the U.S. Food and Drug Administration [7], this by-product possess a high potential to be used as alternative to synthetic antioxidants in food processing such as butylated hydroxytoluene and butylated hydroxyanisole, coinciding with consumers demand for healthy and functional foods with no chemical additives [8]. For this reason, academic and industrial research is focusing on the use of grape pomace powder (GPP) as a food additive or novel ingredient in different food productions [9].

Dairy products contain a low concentration of phenolic compounds, antioxidants and fibers [10]. To this purpose, the fortification of cheeses with non-dairy ingredients represents an improved strategy to enhance the functional and bioactive properties of the final products [11]. So far, the fortification with GPP was proposed for various dairy products obtained with bovine milk such as fermented milk beverages [12], yogurt [13] and processed semi-hard and hard cheeses [14].

Sicily is a region of southern Italy characterized by an intense breed of sheep that play an important role in the protection of the local cultural heritage related to the traditions of shepherds, uses and habits of the mountain populations and typical cuisine. In this region, sheep milk is almost totally processed into traditional cheese types, but several cheese producers pushed research institutes to develop dairy functional foods in order to enlarge the number of dairy products and, especially, to provide cheese with a positive image among consumers aware of the effects of bioactive compounds on the human body.

The objective of the present work was to produce a novel fresh ewes’ milk pressed cheese with the addition of GPP and selected *Lactococcus lactis* strains resistant to the main grape polyphenols [15]. The final cheeses were subjected to the evaluation of the microbiological, physicochemical, sensory and functional aspects.

## 2. Materials and Methods

### 2.1. Grape Pomace Powder Production

Red wine grape pomace of Nero d’Avola cultivar composed by a mix of grape seeds and skins was provided by a winemaking company of Trapani (Southern Italy) at the end of the vintage 2019. Grape pomace was collected after 150 d of post-fermentation maceration. Following the methodology described by Marchiani et al. [14], grape pomace were dried at 54 °C for 48 h in a semi-industrial oven (Compact Combi, Electrolux, Pordenone, Italy). The dry grape pomace was reduced to a particle size of 250 μm through a Retsch centrifugal Mill ZM1 (Haan, Germany).

### 2.2. Strains and Development of Natural Milk Starter Cultures

The strains *Lactococcus lactis* Mise36, Mise94 Mise169 and Mise190, belonging to the culture collection of the Department of Agricultural, Food and Forest Sciences (University of Palermo, Italy), were used in this study. These strains were previously tested for their resistance to GPP and for their main dairy traits [15]. The cultures were individually overnight grown at 30 °C in M17 broth (Biotec, Grosseto, Italy), centrifuged at 10,000× *g* for 5 min to separate the cells from supernatant, washed and re-suspended in Ringer’s solution (Sigma-Aldrich, Milan, Italy). All strains were individually inoculated into 1 L of whole fat UHT milk (Conad, Mantova, Italy) at the final concentration of about 10^6^ CFU/mL obtaining four distinct natural milk starter cultures (NMSC). After incubation for 24 h at 30 °C, the NMSCs were separately used for cheese making.

### 2.3. Experimental Cheese Production and Sample Collection

Cheese productions were carried out under controlled conditions at a dairy pilot plant (Biopek, Gibellina, Italy) using ewes’ milk from the indigenous Sicilian sheep breed “Valle del Belice” during February 2020. The experimental plan included eight different cheese productions as reported in Figure 1. For each strain, 40 L of pasteurized ewes’ milk was divided into two plastic vats (20 L each) representing two different trials. Both vats were inoculated with 200 mL of the corresponding NMSC (Mise36, Mise94, Mise169 and Mise190) to reach a final cell density of 10^7^ CFU/mL. One vat represented the control cheese production (CCP) while the second vat represented the experimental cheese production (ECP) that, after curd extraction, was added with 1% (*w*/*w*) GPP.

Both CCP and ECP were performed applying “primosale” pressed cheese technology as reported in Figure 2.

Cheese productions were carried out in duplicate in two consecutive weeks. The measurement of pH during cheese making (from milk to curd) was carried out with a portable pH-meter pH 70 + DHS (XS Instruments, Carpi, Italy). In order to follow the curd acidification, one sample of curd was collected from each production and kept at ambient temperature for 7 days. Each curd sample was subjected to the monitoring of pH at 2-h intervals for the first 8 h and, then, after 1, 2, 3 and 7 days from milk curdling.

The following matrices were sampled during cheese production: bulk milk, pasteurized bulk milk, inoculated milk after addition of NMSC, curd, GPP and cheese after 1 month of ripening occurred at 13 °C and 85% relative humidity.

### 2.4. Microbiological Analyses

Cell suspensions of milk samples were subjected to decimal serial dilutions in Ringer’s solution (1:10), while GPP, curd and cheese samples were first homogenized in Ringer’s solution by a stomacher (Bag-Mixer 400; Interscience, Saint Nom, France) for 2 min at the maximum speed (blending power 4) and then serially diluted. Cell suspensions of GPP were subjected to plate count for the main microbial groups belonging to the pro-technological, spoilage and pathogenic populations following the approach of Cruciata et al. [16].

Cell suspensions of raw milk and pasteurized milk were analyzed for total mesophilic microorganisms (TMM), mesophilic rods and cocci as reported by Barbaccia et al. [15].

Milk inoculated with each NMSC, curd and cheese samples were analyzed only for the levels of TMM and *L. lactis* on skim milk agar [17] incubated aerobically at 30 °C for 72 h and M17 agar incubated anaerobically at 30 °C for 48 h, respectively. All media and supplements were purchased from Biotec. Plate counts were performed in duplicate.

### 2.5. Phenotypic Grouping, Genotypic Differentiation and Identification of Thermoduric LAB

After growth, all colonies with different morphologies [in order to collect the total lactic acid bacteria (LAB) biodiversity] from the highest dilutions of pasteurized milk sample suspensions were isolated from M17 agar plates and purified by sub-culturing. After microscopic inspection, the pure cultures were tested by KOH assay to determine Gram type and for the presence of catalase by suspension of colonies into H_2_O_2_ 5% (*v*/*v*). All presumptive LAB cultures (Gram-positive and catalase-negative) were grouped on their morphological/physiological/biochemical traits as described by Gaglio et al. [18]. Cell lysis for DNA extraction was performed by using DNA-SORB-B kit (Sacace Biotechnologies Srl, Como, Italy) following the protocol provided by the manufacturer. The differentiation at strain level was performed by randomly amplified polymorphic DNA (RAPD)-PCR analysis as reported by Gaglio et al. [19]. RAPD profiles were analyzed through Gelcompare II software version 6.5 (Applied-Maths, Sin Marten Latem, Belgium). Genotypic identification was performed by sequencing the 16S rRNA gene following the procedures applied by Gaglio et al. [20].

### 2.6. Persistence of the Added Strains

The dominance of the strains individually inoculated as starter cultures (*L. lactis* Mise36, Mise94, Mise169 and Mise190) over LAB resistant to pasteurization was confirmed, after colony isolation, by microscopic inspection and RAPD-PCR profile comparison between LAB collected during cheeses making and those of *L. lactis* Mise36, Mise94, Mise169 and Mise190 pure cultures.

### 2.7. Physicochemical Analysis of Cheeses

Color of external and internal surfaces of the cheeses of the cheeses was assessed by a Minolta Chroma Meter CR300 (Minolta, Osaka, Japan) using the illuminant C; measurements of lightness (L*, from 0 = black, to 100 = white), redness (a*, from red = +a, to green = −a) and yellowness (b*, from yellow = +b, to blue = −b) were performed according to the CIE L* a* b* system [21].

The maximum resistance to compression (compressive stress, N/mm^2^) of samples (2 cm × 2 cm × 2 cm) kept at room temperature (22 °C) was measured, as index of cheese hardness, with an Instron 5564 tester (Instron, Trezzano sul Naviglio, Milan, Italy).

The freeze-dried cheese samples were analyzed for the content of dry matter (DM), fat, protein (N × 6.38) and ash as reported by Bonanno et al. [22].

The products of secondary lipid oxidation were determined as thiobarbituric acid-reactive substances (TBARS), expressed as μg malonylaldehyde (MDA)/kg DM, as reported by Bonanno et al. [22]. Each physicochemical determination was assessed in duplicate.

### 2.8. Volatile Organic Compounds Emitted from Cheeses

Three grams of dried grape pomace and 5 g of chopped cheese samples, were put into 25 mL glass vials sealed with silicon septum. Extraction of volatile compounds were performed through the headspace solid phase microextraction SPME (DVB/CAR/PDMS, 50 mm, Supelco) fiber. The samples were exposed to the fiber under continuous stirring at 60 °C for 15 min. After adsorption, the SPME fiber was thermally desorbed for 1 min through a splitless GC injector at 250 °C. The chromatographic analyses was performed by a gas chromatograph (Agilent 6890) equipped with a mass selective detector (Agilent 5975 c) and a DB-624 capillary column (Agilent Technologies, 60 m, 0.25 mm, 1.40 µm). Chromatographic conditions were as follows: helium carrier gas at 1 mL/min and an oven temperature program with a 5 min isotherm at 40 °C followed by a linear temperature increase of 5 °C min up to 200 °C, where it was held for 2 min. The MS scan conditions applied were: scan acquisition mode; 230 °C Interface temperature; acquisition mass range from 40 to 400. For each sample three replicates were analyzed. The identification of significant volatile compounds were performed through a comparison of the MS spectra with NIST05 library. The relative proportions of the identified constituents were expressed as percentages obtained by GC-MS peak area normalization with total area of significant peaks.

### 2.9. Sensory Evaluation

All cheeses were evaluated by sensory analysis in order to define and detect differences between CCP and ECP. The cheese samples were cubed (approximately 1 cm each side) and then coded and presented on white paperboard plates in a random order. The judges also had available an entire transverse slice of each cheese for evaluating appearance attributes. A total of twelve descriptive attributes were judged by a panel of 11 assessors members (six men and five woman, from 21 to 65 years old). The judges had several years of experience in sensory evaluation of dairy products; however, they were specifically trained for cheese attribute evaluation following the ISO 8589 [23] indications. Each attribute was chosen among those reported by Niro et al. [24] and evaluated by Costa et al. [25]. The intensity of each attribute was quantified using a line scale from 0 to 7 (cm) as reported by Faccia et al. [26].

### 2.10. Simulated Gastrointestinal Digestion

Simulated in vitro human digestion procedure, including the oral, gastric and small intestine phases, was performed three times according to Attanzio et al. [27].

*Oral Phase*. Samples of 15.0 g of cheese were homogenized using a Waring blender (Waring, New Hartford, CT, USA) in 40 mL of a buffered pH 6.8 solution simulating saliva. Artificial saliva, prepared following official pharmacopoeia, contained: NaCl (0.126 g), KC1 (0.964 g) KSCN (0.189 g), KH_2_PO_4_ (0.655 g), urea (0.200 g), Na_2_SO_4_·10H_2_O (0.763 g), NH_4_Cl (0.178 g), CaCl_2_·2H_2_O (0.228 g) and NaHCO_3_ (0.631 g) in 1 L of distilled water. The final pH of the preparations (post-oral digest, PO) ranged between 4.0 and 4.5. An aliquot of 5 mL was stored at −80 °C until analysis.

*Gastric and Small Intestinal Phase*. The sample from the oral phase was acidified at pH 2.0 with HCl, and 8 mg/mL porcine pepsin (3200–4500 units/mg) was added. The sample was transferred in an amber bottle, sealed, and incubated in a shaking (100 rpm) water bath (type M 428-BD, Instruments s.r.l., Bernareggio, Mi, Italy) at 37 °C, for 2 h. Then the reaction mixture was placed on ice, and a 5 mL aliquot was stored at −80 °C (post-gastric digest, PG). The pH of the remaining sample was immediately brought to 7.5 with 0.5 N NaHCO_3_, and 2.4 mg/mL porcine bile extract and 0.4 mg/mL of pancreatin from hog pancreas (amylase activity >100 units/mg) were added to initiate the small intestinal phase of digestion. The amber bottle was sealed and incubated in the shaking water bath for 2 h at 37 °C. At the end of the incubation, 5 mL of the reaction mixture (post-intestinal digest, PI) were stored at −80°C until analysis.

*Preparation of the Bioaccessible Fraction*. The PI digest was centrifuged at 167,000× *g*, for 35 min at 4 °C in a Beckman Optima TLX ultracentrifuge, equipped with an MLA-55 rotor (Beckman Instruments, Inc., Palo Alto, CA, USA), to separate the aqueous fraction (bioaccessible fraction, BF) from particulate material. 

Before analysis, samples from each digestion step were centrifuged at 1500× *g* for 10 min at 4 °C and supernatants were brought at pH 2.0 to stabilize polyphenols.

### 2.11. Total Antioxidant Activity

The total antioxidant activity (TAA) of samples was measured using the ABTS radical cation decolorization assay [28]. ABTS•+ was prepared by reacting ABTS with K_2_S_2_O_4_ [29]. Samples were analyzed in duplicate, at three different dilutions, within the linearity range of the assay. The vitamin E hydro-soluble analog, Trolox, was used as reference antioxidant and results were expressed as micromoles of Trolox equivalents per gram of cheese weight.

### 2.12. Membrane Lipid Peroxidation Assay

Pig’s brain was homogenized in 10 mM phosphate buffer saline, pH 7.4 (PBS) and submitted to centrifugation at 9000× *g* for 20 min at 4 °C. Post-mitochondrial supernatant was then centrifuged at 105,000× *g* for 60 min at 4 °C in a Beckman Optima TLX ultracentrifuge. Microsomal pellet was resuspended in PBS and proteins were determined by the Bio Rad colorimetric method [30]. Microsomes, at 2 mg protein/mL concentration, were pre-incubated for 5 min at 37 °C either in the absence (control) or in the presence of variable amounts of the bioaccessible fraction of cheeses. Lipid oxidation was induced by 20 mM 2,2′-azobis (2-amidino-propane) dihydrochloride (AAPH, Sigma) for 60 min at 37 °C following Attanzio et al. [31]. Oxidized lipid formation was monitored after reaction with thiobarbituric acid (TBA), as TBARS [31]. Prior to sample processing, a calibration analytical curve was prepared at concentrations of 1, 5, 10 and 25 nmol, using tetraethoxypropane (TEP) as the standard. The absorbance was measured using a DU 640 Beckmans pectrophotometer (Beckman, Milan, Italy) at the wavelength of 532 nm. The results were expressed as nmol TBARS/mg protein.

### 2.13. Statistical Analysis

Microbiological data and antioxidant capacity were subjected to One-Way Variance Analysis (ANOVA) using XLStat software version 7.5.2 for Excel (Addinsoft, NY, USA) and the differences between mean were determined by Tukey’s test at *p* < 0.05.

The generalized linear model (GLM) procedure in SAS 9 (Version 9.2, SAS Institute Inc., Campus Drive Cary, NC, USA) was used to analyze physicochemical data of cheeses; the model included the effects of cheese trial (1, 2), treatment (TR) with GPP (control, experimental), starter culture (NMSC: Mise36, Mise94 Mise169, Mise190) and the interaction TR*NMSC. When the effect of NMSC and TR*NMSC resulted significant (*p* ≤ 0.05), means comparisons were performed by the Tukey–Kramer multiple test.

Data on sensory evaluations were tested by a 2-factor analysis of variance (ANOVA), using XLStat software version 2020.3.1 for excel, with judges (i = 1…11) and cheeses (j = 1…8) as fixed factors. Least square means (LSM) were compared using T test (*p* < 0.05).

## 3. Results and Discussion

### 3.1. Acidification Kinetics

The value of pH of pasteurized bulk milk was 6.88, while NMSCs reached values ranging between 4.24–4.28. After the addition of the NWSC, that represent 0.1% cows’ milk in ewe’s milk, bulk milk pH dropped, on average, to 6.71.

The average values of the early acidification process for both control and experimental curds are reported in Figure 3.

According to Tukey’s test, statistical significant differences (*p* < 0.0001) were found between control and experimental trials for all measuring time. In particular, the experimental trials showed values 0.3–0.4 points lower than control trials. These differences are mainly imputable to the presence of organic acids such as tartaric acid, malic acid and citric acid in GPP [14].

### 3.2. Microbiological Analyses

The microbiological counts of GPP did not reveal the presence of any of the microbial groups object of investigation. Mainente et al. [32] assessed that the absence of microorganisms in the GPP is due to the oven-drying treatment performed on grape pomace. The ewes’ milk before pasteurization was characterized by a concentration of TMM of 6.32 Log CFU/mL that is higher than the limit of <500,000 CFU/mL reported by the Commission Regulation (EC) No 853/2004 for raw ewes’ milk. High levels of TMM in raw ewe’s milk before processing into cheese are often detected [33,34]; LAB cocci were found at the same level (10^6^ CFU/mL) of TMM, while LAB rods were one Log unit lower. Similar results were previously reported by Guarcello et al. [35] in raw ewes’ milk used for PDO Pecorino Siciliano cheese production. These results indicated that milking month exerts a limited influence on the microbial load of bulk milk. After pasteurization, TMM and coccus LAB were found at 10^3^ CFU/mL while LAB rods were not detected. These results confirmed what previously reported by Gaglio et al. [36] and Rynne et al. [37] that thermoduric indigenous milk LAB are able to survive the pasteurization process. No statistical significant differences (*p* > 0.05) were found for the levels of microorganisms object of investigation during all steps of cheese making. After inoculation with NMSC, all milks showed approximately 7.0 Log cycles of TMM and almost the same levels of mesophilic coccus LAB, confirming that *L. lactis* inoculums occurred at 10^7^ CFU/mL. After coagulation, all control and experimental curds reached values of TMM and LAB cocci of about 10^8^ CFU/g showing an increase of microbial counts as a consequence of whey draining [34].

These data also confirmed the dominance of lactococci among the microbial community of the curds, reaching values of about 9 Log CFU/g in all control and experimental primosale cheeses. These results highlighted that the addition of 1% (*w*/*w*) of GPP did not influence the fermentation process, carried out by the four strains of *L. lactis* (Mise36, Mise94 Mise169 and Mise190) used individually.

### 3.3. Identification of LAB Resistant to Pasteurization Process

After enumeration, presumptive LAB (Gram-positive and catalase negative) were isolated, purified and analyzed by RAPD-PCR in order to recognize the different strains that overcome the thermal treatment. RAPD analysis showed the presence of 4 different strains (Figure 4a) from a total of 25 presumptive LAB isolates that formed four distinct phenotypical groups 1 for rods and 3 for cocci (Figure 4b).

The sequencing of 16S rRNA gene indicated that the LAB community resistant to pasteurization process was represented by the species *Enterococcus casseliflavus*, *Enterococcus faecalis*, *Enterococcus faecium* and *Lactobacillus fermentum* (recently reclassified as *Limosilactobacillus fermentum*) [38]. All these species are part of the non-starter LAB community implicated in the ripening process of cheeses [39] and represent an essential part of the microbiota of raw ewes’ milk cheeses. In particular, enterococci are often isolated from raw ewe’s milk [40], while *Ls. fermentum* are during cheese ripening [41].

### 3.4. Monitoring of the Added Strains

The development of the added strains was monitored at several cheese production steps collecting 347 isolates which were identified and typed using a polyphasic approach combining microscopic inspection and RAPD-PCR analysis. This approach is commonly applied to monitor the added starter cultures in dairy products [20,42]. Microscopic inspection confirmed that all isolates were cocci with cells organized in short chains, typical of lactococci [15]. The direct comparison of the polymorphic profiles clearly showed the dominance of *L. lactis* Mise36, Mise94, Mise169 and Mise190 both in control and experimental cheeses, excluding any negative influence of GPP.

### 3.5. Physicochemical Analysis of Cheeses

The physical properties and the chemical composition of the cheeses (Table 1) were affected only by the treatment, because no significant variations caused by the starter cultures emerged. Due to the reddish color of GPP, both external and internal surfaces of experimental cheeses showed a low lightness and yellowness and a high redness. However, the indices of internal color recorded in control cheeses were comparable to those measured for primosale cheese after 21 d from production [22], as well as ripened Pecorino cheese [43].

The chemical components of cheese ranged into the levels observed in other investigations for the same cheese typology [22,44]. GPP are poor in lipid components; thus, GPP inclusion in cheese decreased fat level and, as a consequence, protein content increased. The levels of fat in control and experimental cheeses were in the range 44.52–46.31% DM and 39.71–41.83% DM, respectively, while protein content ranged between 43.55% and 46.62% DM in control cheeses and between 47.50% and 50.19% DM in experimental cheeses. A similar behavior was also observed by Marchiani et al. [14] and Frühbauerová et al. [45] in GPP added cow’s cheeses. Moreover, the lower fat content of GPP added cheeses explains their higher hardness, evaluated as resistance to compression, than that registered in control cheeses. 

TBARS values registered for experimental cheeses were higher than those recorded for control cheeses. These results depended on the major oxidation sensitivity of polyunsaturated fatty acids that characterize the lipid profile of GPP [46]. Thus, the antioxidant activity of the phenolic compounds of GPP [14,46] seems not to have preserved cheese from lipid oxidation.

### 3.6. Volatile Organic Compounds Composition of Cheeses

VOC profiles generated by GPP and cheese samples are reported in Table 2. 

The compounds identified belonged to alkanes, aldehydes, monoterpenes, esters, acids, ketones, alcohols and diols. GPP VOC profile was characterized by 20 main compounds and the most abundant belonged to the classes of alcohols, diols and esters groups. The cheeses processed with and without GPP addition emitted 16 and 14 VOCs, respectively with acids, ketones, alcohols and aldehydes being the most represented groups. The higher differences imputable to GPP addition regarded octanoic acid-ethyl ester and 2-phenylethanol. The main acids identified in cheese samples were acetic, hexanoic, butyric and 2-hydroxy4-methyl-pentanoic acids, generally recognized in ewe’s milk cheeses [41,47,48]. Acetic acid may be produced by carbohydrate catabolism by LAB, 2-hydroxy-4-methyl pentanoic acid is formed enzymatically from the corresponding amino acid (L-leucine) [48]. Hexanoic and butyric acids derive mainly from the action of the lamb rennet used for curdling, responsible for the high amounts of short-chain free fatty acids [49]. Free fatty acids (FFA) are responsible for cheese flavor both directly and indirectly. FFA are precursors of odor-active compounds such as methyl ketones, aldehydes, esters and lactones [50]. Even though FAA were present in all cheese samples, esters were poorly detected. No ester compound was identified in control cheeses probably because the ripening period is particularly short [48,51,52]. Two aldehydes (hexenal and heptanal) were detected in cheese with and without GPP addition. In general, cheese VOC profile derives from the hydrolysis or metabolism of carbohydrates, proteins and fats due to the activity of LAB [53]. 2,3 butanediol and 3 hydroxy 2 butanone were found in all cheeses independently on the presence of GPP. Both compounds are generated from the metabolism of carbohydrates (lactose and citrate) by LAB [54]. As a matter of fact, GPP addition contributed to the presence of octanoic acid, ethyl ester and 2-phenylethanol.

### 3.7. Sensory Test

All cheeses were subjected to the sensory analysis and the results are reported in Table 3. 

As reported by Torri et al. [55] the addition of GPP exerts a strong effect on the sensory parameters of dairy products. In this study, except for salt attribute, which was reported not significantly different for judges and cheeses, all other sensory attributes were scored different for cheeses, and not significantly different for judges. In detail, the addition of GPP increased odor and aroma intensity, acid perception, fiber, friability, adhesiveness and humidity, but influenced negatively sweet and hardness. Similar results were observed by Costa et al. [25] and Lucera et al. [56], who tested white and red wine grape pomace to enrich bovine primosale cheese and spreadable cheese, respectively. The overall assessment clearly indicated the cheeses from the trials inoculated with the strain Mise94 (with and without GPP) as those more appreciated.

### 3.8. Antioxidant Properties

Bioactive peptides and phenolic compounds released during digestion of the cheese are considered to be the components primarily responsible for its antioxidative properties [57,58]. GPP is a very rich source of polyphenol compounds with potential health-promoting effects due to the compound’s ability to counteract with body oxidative stress [59]. Contribution of GPP components to the reducing potential of the cheese was estimated while evaluating the antioxidant capacity of GPP-fortified primosale EPC94 compared to the unenriched cheese CPC94, which were found to be the most appreciated by the judges. To simulate the degradation of the matrix in a gastrointestinal environment, samples of both the cheeses were submitted to in vitro digestion and the Total Antioxidant Activity (TAA) in the different digestion phases was measured by the ABTS^+●^ decolorization assay. As shown in Figure 5, although post-oral fractions of both the cheeses showed reducing activity not significantly different, simulated PG fraction of EPC94 had a TAA value (0.342 ± 0.028 µmol TE/g) higher (*p* < 0.001) than that of the CPC94 cheese (0.234 ± 0.019 µmol TE/g). It is plausible that digestion of casein micelles by gastric pepsin has solubilized the incorporated polyphenols in GPP-enriched cheese, resulting in an increase of the reducing activity of the fraction [60]. After intestinal digestion, antioxidant activity of the fractions from both cheeses was about 50% higher than that measured in the relevant gastric digesta, possibly because of release of antioxidant fat-soluble vitamins or amino acids from the dairy matrix [61,62]. Finally, reducing compounds appeared entirely portioned in the BF, i.e., the soluble fraction of the intestinal digesta available for the absorption, and antioxidant capacity of BF from GPP-enriched cheese (0.590 ± 0.033 µmol TE/g) were 60% higher than that of the unenriched cheese (0.371 ± 0.029 µmol TE/g) (Figure 5).

The antioxidant potential of the bioaccesible fractions of the cheeses was also assessed utilizing an in vitro model of membrane lipid oxidation. Lipid oxidation was induced in bovine brain microsomes (2 mg protein/mL) by AAPH-derived peroxyl radicals (20 mM) and oxidized lipids were spectrophotometrically measured as TBARS. In these conditions, after 60 min incubation at 37 °C, an amount of 0.91 ± 0.07 nmoles TBARS /mg protein was detected (control, Figure 6). When BF from 0.1 g or 0.2 g of CPC94 cheese was added to the microsomal preparation before AAPH, slightly lower amounts of TBARS were measured (0.82 ± 0.06 and 0.75 ± 0.05 nmoles/mg protein, respectively). Interestingly, in the presence of BF from 0.1 g or 0.2 g GPP-fortified primosale EPC94, much more marked dose-dependent inhibition of the lipid oxidation was evident (0.55 ± 0.04 and 0.29 ± 0.01 nmoles TBARS/mg protein, respectively) (Figure 6). 

Collectively our results demonstrate that polyphenols from GPP added into primosale cheese significantly boost antioxidant properties of the product, conferring to it potential capacity to control oxidative stress. As reported by Ianni and Martino [63] winemaking by-products enhanced the antioxidant capacity of several beverages, but to the best of our knowledge, no previous work evaluated this aspects in GPP-enriched cheeses. 

## 4. Conclusions

The investigation on GPP-enriched ovine primosale cheese revealed that winemaking by-products did not alter the microbiological parameters during the ripening carried out with *L. lactis*. The chemical composition of the final cheeses clearly showed that the enrichment with GPP decreased the fat content and increased the protein content as well as the values of secondary lipid oxidation. GPP addition impacted cheese VOC profiles with 2-phenylethanol and octanoinc acid, ethyl ester. The sensory analysis evidenced the highest overall acceptability for the cheeses produced with the strain MISE94 (with and without GPP). From the functional point of view, GPP addition increased antioxidant activity of the cheese after that the dairy matrix was degraded by simulated digestive process. Further studies will be carried out for a more accurate validation of this manufacturing method for ovine cheeses considering also the addition of selected LAB in multi-strain combination.

## Figures and Tables

**Figure 1 foods-10-00461-f001:**
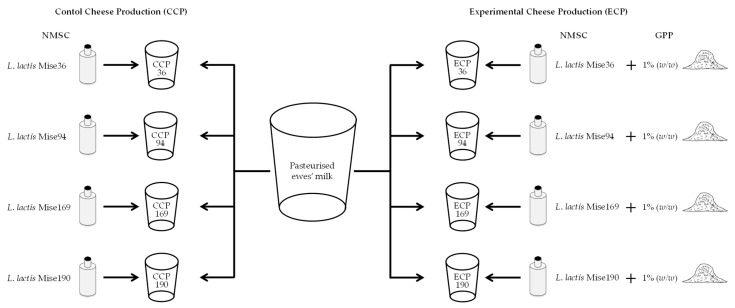
Experimental design of cheese productions. Abbreviations: NMSC, natural milk starter culture; GPP, grape pomace powder; *L. Lactococcus*.

**Figure 2 foods-10-00461-f002:**
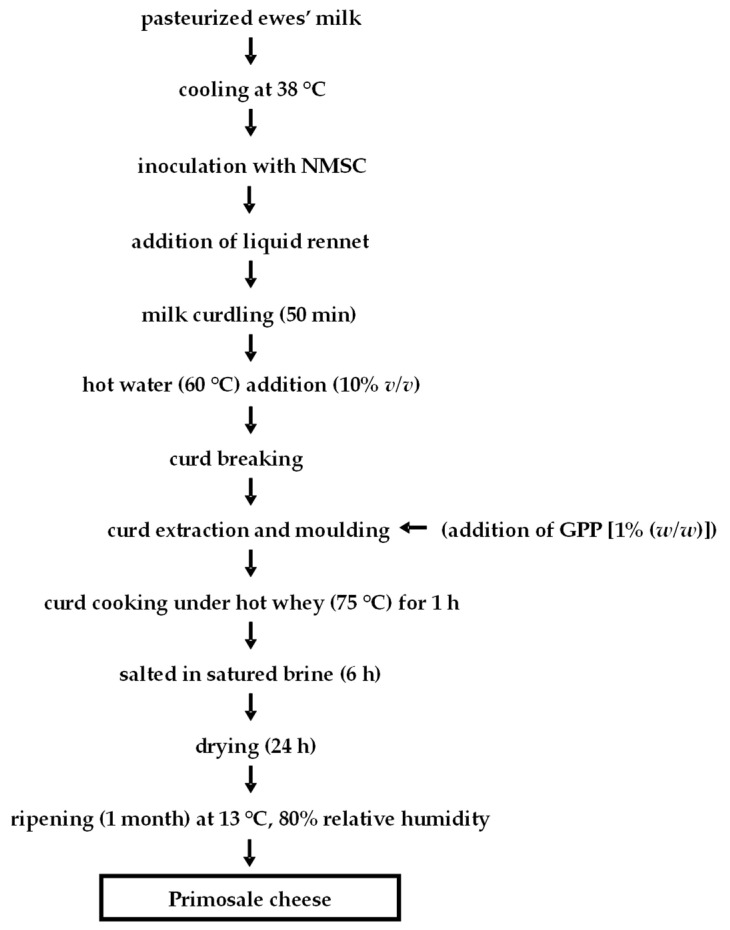
Flow diagrams of Primosale cheese production. Abbreviations: NMSC, natural milk starter culture; GPP, grape pomace powder.

**Figure 3 foods-10-00461-f003:**
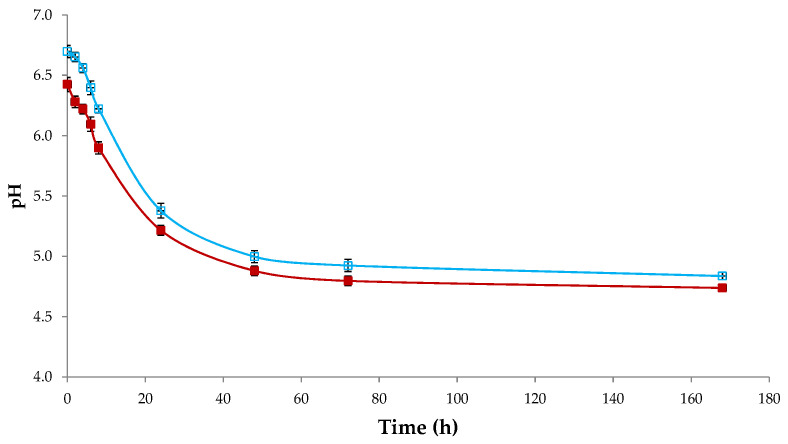
pH decrease during 7-days of curd acidification. Empty symbols: control curds. Full symbols: experimental curds. Results indicate mean values ± SD of the four determinations (carried out in duplicate for two independent productions) of all trials.

**Figure 4 foods-10-00461-f004:**
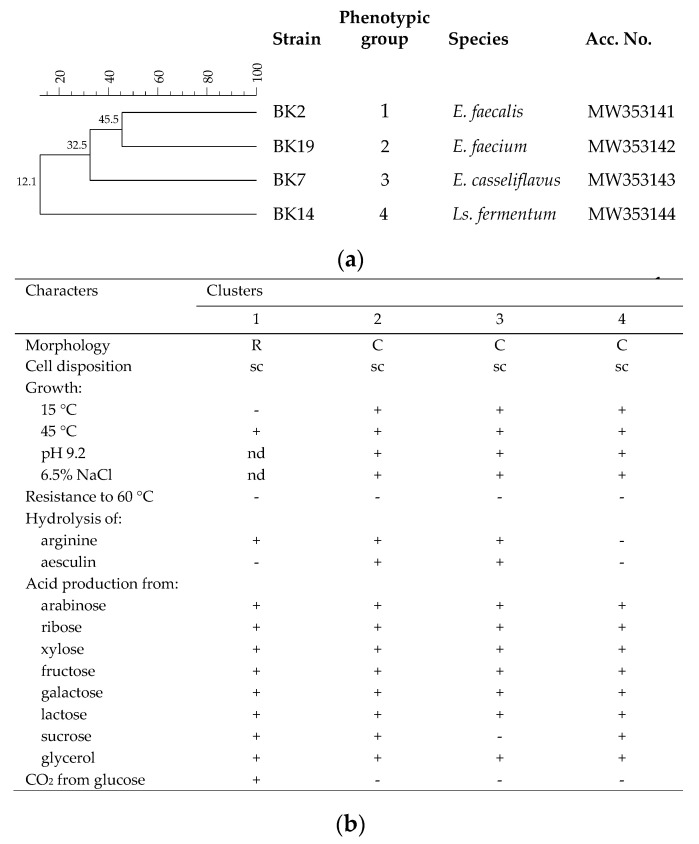
Differentiation of lactic acid bacteria (LAB) isolates from pasteurized ewes’ milk. (**a**) Dendrogram obtained with combined randomly amplified polymorphic DNA (RAPD)-PCR patterns of the LAB strains identified; (**b**) phenotypic grouping of the LAB isolates based of morphological, physiological and biochemical traits. Abbreviations: *E.*, *Enterococcus*; *Ls.*, *Limosilactobacillus*; R, rod; C, coccus; s.c., short chain; n.d., not determined.

**Figure 5 foods-10-00461-f005:**
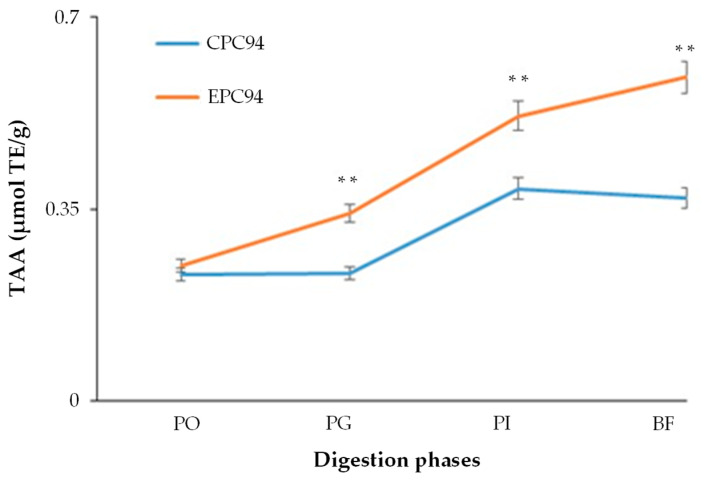
Antioxidant activity of cheeses during in vitro simulated human digestion. In vitro digestion conditions and measurement of total antioxidant activity (TAA) of the different digestion phases. Within the same digestion phase, values are significantly different with ** *p* < 0.001. Abbreviations: PO, post-oral digest; PG, post-gastric digest; PI, post-intestinal digest; BF, bioaccessible fraction; CPC190, control primosale cheese with *L. lactis* MISE94; EPC190 experimental primosale cheese with 1% of GPP and *L. lactis* MISE94.

**Figure 6 foods-10-00461-f006:**
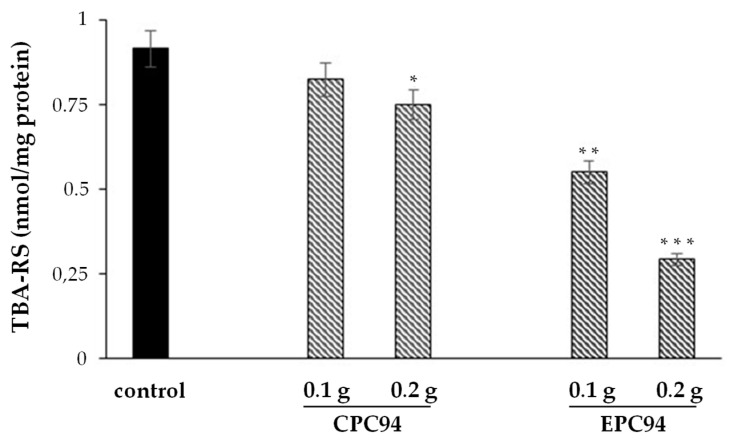
Thiobarbituric acid-reactive substances (TBARS) formation after AAPH-induced oxidation of microsomal membrane either in the absence (control) or in the presence of bioaccessible fraction obtained following simulated human digestion of cheeses. Microsomes, at 2 mg of protein per mL of reaction mixture, were incubated at 37 °C in the presence of 20 mM AAPH for 60 min. TBARS was spectrophotometrically measured as reported in the methods. Value are the mean ± SD of three determinations performed in duplicate. In comparison to the control, values are significantly different with * *p* < 0.05, ** *p* < 0.001; *** *p* < 0.0001 (Student’s *t*-test). Abbreviations: CPC190, control primosale cheese with *L. lactis* MISE94; EPC190 experimental primosale cheese with 1% of GPP and *L. lactis* MISE94.

**Table 1 foods-10-00461-t001:** Physicochemical traits of primosale cheeses at 1-month of ripening.

		Treatment (TR)		Starter Culture (NMSC)				SEM	Significance *p* <		
				MISE36	MISE94	MISE169	MISE190		TR	NMSC	TR*NMSC
External color	lightness (L*)	Control	62.24	62.17	61.15	60.96	64.70	2.34	<0.0001	0.4294	0.3086
		Experimental	31.40	26.93	33.39	33.21	32.07				
		Total ^a^		44.55	47.27	47.08	48.38				
	redness (a*)	Control	−5.82	−5.82	−5.99	−5.79	−5.68	0.26	<0.0001	0.5537	0.6203
		Experimental	1.77	2.08	1.71	1.43	1.86				
		Total		−1.87	−2.14	−2.18	−1.91				
	yellowness (b*)	Control	13.64	13.57	14.04	12.54	14.40	0.47	<0.0001	0.1419	0.3863
		Experimental	1.82	1.73	1.89	1.67	1.97				
		Total		7.65	7.97	7.11	8.19				
Internal color	lightness (L*)	Control	70.39	71.31	70.47	69.16	70.61	1.46	<0.0001	0.3541	0.7342
		Experimental	43.22	42.70	43.72	41.35	45.13				
		Total		57.00	57.10	55.25	57.87				
	redness (a*)	Control	−4.90	−4.95	−5.20	−4.62	−4.82	0.36	<0.0001	0.2181	0.6629
		Experimental	3.45	3.12	3.37	4.13	3.16				
		Total		−0.91	−0.91	−0.25	−0.83				
	yellowness (b*)	Control	12.60	13.04	13.82	11.53	12.01	0.68	<0.0001	0.2068	0.4600
		Experimental	3.67	4.09	3.61	3.27	3.69				
		Total		8.57	8.71	7.40	7.85				
Hardness, N/mm^2^		Control	1.03	1.01	0.98	1.10	1.02	0.048	<0.0001	0.0906	0.4099
		Experimental	1.23	1.31	1.15	1.26	1.19				
		Total		1.16	1.06	1.18	1.11				
Chemical composition	Dry matter (DM), %	Control	63.23	63.19	62.75	63.46	63.51	0.48	0.2497	0.8107	0.6814
		Experimental	63.65	64.05	63.59	63.49	63.49				
		Total		63.62	63.17	63.47	63.50				
	Ash, % DM	Control	6.37	6.32	6.22	6.51	6.45	0.21	0.5420	0.8912	0.7650
		Experimental	6.47	6.62	6.43	6.43	6.40				
		Total		6.47	6.33	6.47	6.42				
	Protein, % DM	Control	45.45	45.43	43.55	46.62	46.61	1.44	0.0147	0.3339	0.7330
		Experimental	48.72	49.56	47.64	50.19	47.50				
		Total		47.49	45.59	48.40	46.85				
	Fat, % DM	Control	45.69	46.31	46.06	44.52	45.87	1.03	0.0002	0.2815	0.6349
		Experimental	40.63	39.71	41.64	39.35	41.83				
		Total		43.01	43.85	41.93	43.85				
TBARS, μg MDA/kg DM		Control	31.49	33.30	31.47	31.25	29.94	3.8123	0.0173	0.5980	0.4708
		Experimental	38.41	39.46	37.45	32.98	43.75				
		Total		36.38	34.46	32.11	36.85				

^a^ Total means of starter cultures. Abbreviations: SEM, standard error of mean; TBARS, thiobarbituric acid-reactive substances; MDA, malonylaldehyde.

**Table 2 foods-10-00461-t002:** Volatile organic compounds emitted from GPP and primosale cheeses at 1-month of ripening.

Chemical Compounds ^a^	Samples								
	GPP	CPC36	EPC36	CPC94	EPC94	CPC169	EPC169	CPC190	EPC190
Acids									
Acetic acid	n.d.	7.4	15.6	13.6	16.8	12.6	12.6	11.1	13.9
Butanoic acid	n.d.	4.1	10.6	12.5	7.3	7.3	7.3	6.0	9.9
4-Hydroxybutanoic acid	4.1	n.d.	n.d.	n.d.	n.d.	n.d.	n.d.	n.d.	n.d.
Hexanoic acid	1.6	1.9	4.6	5.7	3.4	5.8	4.2	3.6	4.8
Pentanoinc acid-2-hydroxy-4-methyl	n.d.	1.7	3.4	1.4	1.9	1.4	1.2	1.2	1.2
Nonanoic acid	3.4	n.d.	n.d.	n.d.	n.d.	n.d.	n.d.	n.d.	n.d.
ketones									
2-Pentanone	n.d.	2.2	0.8	1.5	0.9	2.7	0.8	2.7	0.8
3-Hydroxy-2-butanone	n.d.	23.8	41.8	31.1	49.6	7.6	22.3	14.0	34.6
2-Heptanone	n.d.	0.2	0.4	0.8	0.4	0.5	0.6	1.3	0.3
p-Phenylacetophenone	4.2	n.d.	n.d.	n.d.	n.d.	n.d.	n.d.	n.d.	n.d.
Alcohol									
Isoamyl alcohol	4.9	7.8	2.7	6.7	8.2	27.2	37.7	10.4	15.0
2-Butanol	n.d.	0.7	2.6	4.8	2.1	2.5	2.6	1.7	4.1
2-Phenylethanol	11.3	n.d.	1.4	n.d.	1.0	n.d.	1.5	n.d.	0.9
Hydrocarbons									
Hexane 2-methyl	n.d.	3.0	0.7	1.0	0.6	2.2	0.7	2.6	0.8
Heptane 2,4-dimethyl	3.2	2.0	2.7	3.0	3.2	2.6	1.4	1.6	3.2
Nonane	2.2	n.d.	n.d.	n.d.	n.d.	n.d.	n.d.	n.d.	n.d.
Nonane 2,5-methyl	2.3	n.d.	n.d.	n.d.	n.d.	n.d.	n.d.	n.d.	n.d.
Decane	1.8	n.d.	n.d.	n.d.	n.d.	n.d.	n.d.	n.d.	n.d.
Dodecane	2.3	n.d.	n.d.	n.d.	n.d.	n.d.	n.d.	n.d.	n.d.
Hexadecane	1.7	n.d.	n.d.	n.d.	n.d.	n.d.	n.d.	n.d.	n.d.
Aldeyde									
Hexanal	3.2	22.0	2.8	5.6	0.2	18.1	1.0	15.8	2.4
Heptanal	n.d.	21.8	4.4	6.6	0.5	6.6	1.5	25.0	2.6
Nonanal	1.7	n.d.	n.d.	n.d.	n.d.	n.d.	n.d.	n.d.	n.d.
Monoterpene									
D-Limonene	6.3	n.d.	n.d.	n.d.	n.d.	n.d.	n.d.	n.d.	n.d.
α-Pinene	2.1	n.d.	n.d.	n.d.	n.d.	n.d.	n.d.	n.d.	n.d.
Carene	1.5	n.d.	n.d.	n.d.	n.d.	n.d.	n.d.	n.d.	n.d.
Esters									
Octanoinc acid, ethyl ester	9.6	n.d.	1.3	n.d.	1.1	n.d.	1.2	n.d.	0.7
Butanedioic acid, diethyl ester	2.2	n.d.	n.d.	n.d.	n.d.	n.d.	n.d.	n.d.	n.d.
Decanoic acid, ethyl ester	9.7	n.d.	n.d.	n.d.	n.d.	n.d.	n.d.	n.d.	n.d.
Diol									
2,3-Butanediol	20.6	1.4	4.1	5.6	2.9	2.9	3.4	2.8	4.9

^a^ Data are means percentage of three replicate expressed as (peak area of each compound/total area of significant peaks) × 100. Abbreviations: GPP, grape pomace powder; CPC36, CPC94, CPC169 and CPC190, control primosale cheese with *L. lactis* MISE36, MISE94, MISE169 and MISE190, respectively; EPC36, EPC94, EPC169 and EPC190 experimental primosale cheese with 1% of GPP and *L. lactis* MISE36, MISE94, MISE169 and MISE190, respectively; n.d., not detectable.

**Table 3 foods-10-00461-t003:** Evaluation of the sensory attributes of primosale cheeses at 1-month of ripening.

Attributes	Trial								SEM	*p*-Value	
	CPC36	EPC36	CPC94	EPC94	CPC169	EPC169	CPC190	EPC190		Judges	Cheeses
Intensity of odor	5.32 ^cd^	5.94 ^bc^	6.22 ^b^	6.78 ^a^	5.06 ^d^	5.76 ^bcd^	5.26 ^cd^	5.91 ^bc^	0.07	0.053	<0.0001
Intensity of aroma	5.41 ^cde^	5.95 ^b^	5.99 ^b^	6.55 ^a^	4.95 ^e^	5.46 ^cd^	5.16 ^de^	5.82 ^bc^	0.06	0.099	<0.0001
Sweet	5.02 ^b^	4.47 ^c^	5.57 ^a^	5.05 ^b^	5.19 ^b^	4.71 ^c^	5.15 ^b^	4.62 ^c^	0.04	0.627	<0.0001
Salt	3.53 ^a^	3.48 ^a^	3.43 ^a^	3.46 ^a^	3.47 ^a^	3.45 ^a^	3.44 ^a^	3.46 ^a^	0.03	0.999	0.999
Acid	2.42 ^b^	3.28 ^a^	2.44 ^b^	3.22 ^a^	2.37 ^b^	3.20 ^a^	2.48 ^b^	3.32 ^a^	0.05	0.733	<0.0001
Astringent	0.00 ^b^	1.66 ^a^	0.00 ^b^	1.64 ^a^	0.00 ^b^	1.60 ^a^	0.00 ^b^	1.59 ^a^	0.08	0.999	<0.0001
Friability	1.53 ^b^	2.42 ^a^	1.50 ^b^	2.46 ^a^	1.42 ^b^	2.31 ^a^	1.56 ^b^	2.33 ^a^	0.05	0.860	<0.0001
Fiber	1.39 ^b^	2.56 ^a^	1.30 ^b^	2.48 ^a^	1.36 ^b^	2.56 ^a^	1.42 ^b^	2.58 ^a^	0.06	0.952	<0.0001
Adhesiveness	2.41 ^b^	3.49 ^a^	2.45 ^b^	3.54 ^a^	2.38 ^b^	3.42 ^a^	2.46 ^b^	3.44 ^a^	0.05	0.998	<0.0001
Hardness	4.18 ^a^	2.45 ^b^	4.05 ^a^	2.56 ^b^	4.02 ^a^	2.46 ^b^	4.06 ^a^	2.38 ^b^	0.08	0.985	<0.0001
Humidity	2.53 ^b^	3.62 ^a^	2.40 ^b^	3.51 ^a^	2.28 ^b^	3.56 ^a^	2.60 ^b^	3.65 ^a^	0.06	0.971	<0.0001
Overall assessment	4.36 ^c^	4.56 ^c^	5.74 ^b^	6.07 ^a^	4.33 ^c^	4.36 ^c^	4.46 ^c^	4.62 ^c^	0.06	0.999	<0.0001

Results indicate mean value. Data within a line followed by the same letter are not significantly different according to Tukey’s test. Abbreviations: CPC36, CPC94, CPC169 and CPC190, control primosale cheese with *L. lactis* MISE36, MISE94, MISE169 and MISE190, respectively; EPC36, EPC94, EPC169 and EPC190 experimental primosale cheese with 1% of GPP and *L. lactis* MISE36, MISE94, MISE169 and MISE190, respectively.

## Data Availability

All data included in this study are available upon request by contacting the corresponding author.

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
