# Peer review of "The Use of Winery by-Products to Enhance the Functional Aspects of the Fresh Ovine “Primosale” Cheese"

_foods, 2021, doi:10.3390/foods10020461_

Round 1
Reviewer 1 Report
Abstract ECP94????
Lines 34-35. Delete
Line 37. “a mix of grape skins, seeds and stalks”. I do not agree with the stalks. Usually grape pomace is only composed of grape skins and seeds. The authors should be very careful since they also use the same term “grape pomace” but in their case is only for grape skin and seed.
Line 37 Winery by-products line 38 grape pomace. Please use only one this is very misleading.
Line 68. Please provide (better as table) the average composition of the grape pomace used (sugars, fat, protein, total phenolic content, antioxidant activity etc). In addition the authors should provide details about the production of this pomace (is it after the completion of wine making?)
Lines 76-68. Please revise. No meaning.
Line 92. “individual starter L. lactis culture (200 mL)” Please explain. What was the pH of that? This will not affect the pH of the cheese?
Line 92. Therefore you have cow milk UHT (0.1%) in ewe’s milk.
Figure 2. Please explain hot water addition. Quantity?
Figure 2. Correct drying.
Line 145. “External and internal colour”. Please explain.
Line 153. Freeze-drying?? Provide details
Line 177. Is this panel appropriate? Were thy trained? Please provide details.
Line 212. Please explain the use of only one method ABTS. It is a usual practice to use a second method to evaluate antioxidant activity.
Figure 3. Please delete. There is no information. There is not significant differences and the results could be easily presented in the text.
Table 1. Please explain “total”. In addition these results are for which day?
Table 2. Results for which day?
Table 2. I cannot understand the presence of GPP in cheeses. Please delete column GPP. There is nothing to compare. In addition if the authors remove GPP there are only few volatiles detected.
In my opinion the VOC analysis should be removed. There is nothing to add here. There is no conclusions and I am not sure if this was contacted correctly.
Table 3. I would like to ask the authors if during sensory evaluation the judges were able to see the samples of there was a blind test.
Figure 5. Please explain how is it possible the two samples to have the same initial antioxidant capacity.
Where are analyses like pH, acidity, etc?
Discussion using similar references is missing. Please consider the following
Frühbauerová, M., Červenka, L., Hájek, T., Salek, R. N., Velichová, H., & Buňka, F. (2020). Antioxidant properties of processed cheese spread after freeze-dried and oven-dried grape skin powder addition. Potravinarstvo Slovak Journal of Food Sciences, 14, 230-238.
Lucera, A., Costa, C., Marinelli, V., Saccotelli, M. A., Del Nobile, M. A., & Conte, A. (2018). Fruit and vegetable by-products to fortify spreadable cheese. Antioxidants, 7(5), 61.
Torri, L., Piochi, M., Marchiani, R., Zeppa, G., Dinnella, C., & Monteleone, E. (2016). A sensory-and consumer-based approach to optimize cheese enrichment with grape skin powders. Journal of Dairy Science, 99(1), 194-204.
Lines 474-476. This link does not work. In addition the authors should provide the latest date that they reached this website 2020-2021…
Author Response
Answers to Reviewer 1:
- Thanks for your comments. All your suggestions were considered and the changes in the text were highlighted in yellow.
Abstract ECP94????
- ECP94 is the abbreviation for the experimental cheese produced using as fermenting agent the strain of Lactococcus lactis Mise94. The abstract has been modified for higher clarity (L17,24).
Lines 34-35. Delete
- Deleted.
Line 37. “a mix of grape skins, seeds and stalks”. I do not agree with the stalks. Usually grape pomace is only composed of grape skins and seeds. The authors should be very careful since they also use the same term “grape pomace” but in their case is only for grape skin and seed.
- We apologize for the mistake. The term stalks has been deleted (L35).
Line 37 Winery by-products line 38 grape pomace. Please use only one this is very misleading.
- The sentence has been modified to address your request (L35-36).
Line 68. Please provide (better as table) the average composition of the grape pomace used (sugars, fat, protein, total phenolic content, antioxidant activity etc). In addition the authors should provide details about the production of this pomace (is it after the completion of wine making?)
- Thanks for this suggestion. This is a quite interesting aspect to be deepened in a specific paper aimed to evaluate GPP from different wine grape cultivars. In the present work, we mainly focused on the final cheese aspects (lipid oxidation) rather than polyphenols, but it will be the subject of a future work to be prepared very soon.
Lines 76-78. Please revise. No meaning.
- Revised (L107-110).
Line 92. “individual starter L. lactis culture (200 mL)” Please explain. What was the pH of that? This will not affect the pH of the cheese?
- We agree with your comment and the sentence was modified to address your request (L123-124). However, in order to provide clarity, in our study four strain of Lactococcus lactis (Mise36, Mise94, Mise169 and Mise190) previously selected for their ability to growth in presence of the main grape polyphenols was firstly used individually to produce Natural Milk Starter Culture (NMSC), which secondly were inoculated at a final cell density of 107 CFU/mL in order to drive the acidification process of each trial. Regarding the pH values of NMSC have been added in the text in M&M (L161-165) and Results sections (L323-325).
Line 92. Therefore you have cow milk UHT (0.1%) in ewe’s milk.
- This information has been added in the text (L324-325). However, in order to industrialize GPP-enriched cheese production process, works are being prepared to specifically propagate NMSC in appropriate ewe’s UHT milk and to evaluate volatile differences of the final products.
Figure 2. Please explain hot water addition. Quantity?
- The hot water is added at 10% (v/v) during the breaking of the curd to facilitate syneresis. The quantity has been added in figure 2.
Figure 2. Correct drying.
- Corrected.
Line 145. “External and internal colour”. Please explain.
- The expression has been corrected as “Colour of external and internal surfaces of the cheeses…” (L207).
Line 153. Freeze-drying?? Provide details
- Freeze-drying of cheese samples is performed in several experiments since permits to eliminate the most water amount before analysis without compromise their components. No specific details can be reported since the process changes only in relation to its duration that in turn is linked to the samples humidity.
Line 177. Is this panel appropriate? Were thy trained? Please provide details.
- The panel test has been performed following the ISO 8589 (2007) and the each attribute was evaluate as reported by Costa et al. (2018) who tested white and red wine grape pomace to enrich bovine primosale cheese. Regarding the judges included in the panel had several years of experience in sensory evaluation, furthermore they were trained in a specific session. The sentence has been modified to address your request (L248-250).
Line 212. Please explain the use of only one method ABTS. It is a usual practice to use a second method to evaluate antioxidant activity.
- AU. We also measured DPPH free radical scavenging activity in ethanol solution according Brand-Williams et al. (Use of free radical method to evaluate antioxidant activity. Lebensm. Wiss.Technol., 1995, 28,25–30). The main phenolic compounds in GPP are flavan-3-ols, phenolic acids, (+)-catechins and proanthocyanidins with low solubility in organic solution. Thus we obtained cloudy solutions when adding the samples containing the GPP enriched cheese. So, we decided to carry out only the ABTS radical cation decolorization assay in phosphate saline buffer.
Figure 3. Please delete. There is no information. There is not significant differences and the results could be easily presented in the text.
- Figure 3 has been deled. The results of statistical analysis has been added in the text (L385-387).
Table 1. Please explain “total”. In addition these results are for which day?
- “Total” is referred to the total means of starter cultures, regardless of the GPP inclusion. This information has been provided (L455). All determinations were performed on cheeses after 1-month ripening; this specific information has been added in the titles, also in the tables 2 and 3.
Table 2. Results for which day?
- All determinations were performed on cheeses after 1-month ripening; please see previous answer also.
Table 2. I cannot understand the presence of GPP in cheeses. Please delete column GPP. There is nothing to compare. In addition if the authors remove GPP there are only few volatiles detected.
In my opinion the VOC analysis should be removed. There is nothing to add here. There is no conclusions and I am not sure if this was contacted correctly.
- Thanks for this suggestion, but we performed VOCs analysis, since it is an appropriate information to be provided whit novel foods. GPP was analysed to evaluate the impact of this addition on the new cheese. It is true that GPP do not impact greatly cheese profile, but without this information shown it cannot be simply supposed. As a matter of fact, 2-Phenylethanol and Octanoinc acid, ethyl ester are present in GPP-enriched cheeses and without GPP column is difficult to explain. We provided conclusions regarding this point (L608-609).
Table 3. I would like to ask the authors if during sensory evaluation the judges were able to see the samples of there was a blind test.
- We agree with this observation and for this reason the section of “sensory evaluation” has been improved to address your request (L244-247).
Figure 5. Please explain how is it possible the two samples to have the same initial antioxidant capacity.
- We think that the oral digestion phase is not able to solubilize phenolic compounds of GPP which are mainly linked to the casein micelles, as expressed in lines 431-433. Then in the oral phase only the reducing activity of milk components was measured. Since CPC94 and EPC94 are made from the same milk, it is not surprising that their TAA values are quite similar.
Where are analyses like pH, acidity, etc?
- Acidification was followed by pH evolution. To address your request a new paragraph has been added in M&M (L161-165) and results and discussion section (L326-336). A new picture was provided (Figure 3).
Discussion using similar references is missing. Please consider the following
Frühbauerová, M., Červenka, L., Hájek, T., Salek, R. N., Velichová, H., & Buňka, F. (2020). Antioxidant properties of processed cheese spread after freeze-dried and oven-dried grape skin powder addition. Potravinarstvo Slovak Journal of Food Sciences, 14, 230-238.
Lucera, A., Costa, C., Marinelli, V., Saccotelli, M. A., Del Nobile, M. A., & Conte, A. (2018). Fruit and vegetable by-products to fortify spreadable cheese. Antioxidants, 7(5), 61.
Torri, L., Piochi, M., Marchiani, R., Zeppa, G., Dinnella, C., & Monteleone, E. (2016). A sensory-and consumer-based approach to optimize cheese enrichment with grape skin powders. Journal of Dairy Science, 99(1), 194-204.
- The missing citations have been added in the text (L442, 501-502, 507-508,) and in the references list (L754-755, 778-779, 780-781).
Lines 474-476. This link does not work. In addition the authors should provide the latest date that they reached this website 2020-2021…
- We apologize for the mistake. A new link and the date of the access have been added (L657-658).
The paper deals with a very topical topic - the use of production waste in food technology. Work brings new content to food science. After making small corrections, I recommend the article for publication.
AU. Thanks a lot for your positive comment.

Reviewer 2 Report
Comments:
This manuscript “The Use of Winery by-Products to Enhance the Functional Aspects of the Fresh Ovine “Primosale” Cheese” investigated the influence of the inclusion of grape pomaces powder into the process of making fresh ovine cheese “primosale”.
Recently some articles have been written about similar subjects:
Lucera, A.; Costa, C.; Marinelli, V.; Saccotelli, M.A.; Del Nobile, M.A.; Conte, A. Fruit, and Vegetable By-Products to Fortify Spreadable Cheese. Antioxidants 2018, 7, 61. https://doi.org/10.3390/antiox7050061
García‐Lomillo, J. and González‐SanJosé, M.L. (2017), Applications of Wine Pomace in the Food Industry: Approaches and Functions. COMPREHENSIVE REVIEWS IN FOOD SCIENCE AND FOOD SAFETY, 16: 3-22. https://doi.org/10.1111/1541-4337.12238
So, the topic is not new but, the article, contributes to positive and interesting new data of the topic in question and the introduction summarizes the previous works. Proper experiments were conducted, and the data obtained is remarkably interesting and relevant, not only for the dairy industry but also for the wine and wine byproducts industry.
Author Response
Answers to Reviewer 2:
This manuscript “The Use of Winery by-Products to Enhance the Functional Aspects of the Fresh Ovine “Primosale” Cheese” investigated the influence of the inclusion of grape pomaces powder into the process of making fresh ovine cheese “primosale”.
Recently some articles have been written about similar subjects:
Lucera, A.; Costa, C.; Marinelli, V.; Saccotelli, M.A.; Del Nobile, M.A.; Conte, A. Fruit, and Vegetable By-Products to Fortify Spreadable Cheese. Antioxidants 2018, 7, 61. https://doi.org/10.3390/antiox7050061
García‐Lomillo, J. and González‐SanJosé, M.L. (2017), Applications of Wine Pomace in the Food Industry: Approaches and Functions. COMPREHENSIVE REVIEWS IN FOOD SCIENCE AND FOOD SAFETY, 16: 3-22. https://doi.org/10.1111/1541-4337.12238
So, the topic is not new but, the article, contributes to positive and interesting new data of the topic in question and the introduction summarizes the previous works. Proper experiments were conducted, and the data obtained is remarkably interesting and relevant, not only for the dairy industry but also for the wine and wine byproducts industry.
- Thanks a lot for your positive comments.

Reviewer 3 Report
This article examines the impact of including wine making by products on the composition and quality of cheese also investigating potential benefits of their inclusion on the oxidative potential of the final products.
I found this to be a very nice study. Some comments are listed below.
Abstract
L26 - very higher? significant or not significant.
L28 - its not made clear to the reader why GPP is being added does it have some beneficial properties or is it a case of sustainable nutrition and utilising otherwise waste materials?
Introduction
L40 - on human health.
L41 - diseases related to oxidative stress, examples?
L44 can you provide example of synthetic anti oxidants?
L44 encountering - coinciding with consumers demand for...
L48 Here and elsewhere there are some grammar issues to consider. - do you mean they are denatured by heat treatment? Even the relatively gentle pasteurisation 72 degC x 15s?
L51 - improved stategy
L54 - if it has been done in cheese before what is novel here?
L66 removed second [.]
L76 why were you adding polyphenols is this separate to the prepared GPP?
L124 - why was it necessary to genotype the different strains?
L180 - were these trained assessors?
L220 - do you have a reference for this method?
Section 3.2 can you provide greater clarity as to what this has to do with GPP inclusion or is there any link? This is not routine analysis?
Other than the type of cheese chosen what is the big difference between this study and that by Marchiani 2016?
Section 3.7 - grammar in this section needs to be reviewed.
Is this in vitro model of membrane lipid oxidation typically used?
L419 - very more? Significantly?
L431 - are there any comparison with existing literature how this compares with other foodstuffs or polyphenol enriched foods?
L436 - i think ripening would be a more accurate terminology.
Author Response
Answers to Reviewer 3:
This article examines the impact of including wine making by products on the composition and quality of cheese also investigating potential benefits of their inclusion on the oxidative potential of the final products.
I found this to be a very nice study. Some comments are listed below.
- Thanks for your comments. All your suggestions were considered and the changes in the text were highlighted in light green.
Abstract
L26 - very higher? significant or not significant.
- Modified (L26).
L28 - its not made clear to the reader why GPP is being added does it have some beneficial properties or is it a case of sustainable nutrition and utilising otherwise waste materials?
- We agree and for this reason the sentence has been modified to address your request (L27-29).
Introduction
L40 - on human health.
- Modified (L38).
L41 - diseases related to oxidative stress, examples?
- Added (L39-40).
L44 can you provide example of synthetic antioxidants?
- Added (L43-44).
L44 encountering - coinciding with consumers demand for...
- Modified (L44).
L48 Here and elsewhere there are some grammar issues to consider. - do you mean they are denatured by heat treatment? Even the relatively gentle pasteurisation 72 degC x 15s?
- We apologize for the mistake. The sentence has been rewritten (L80-81) and the consideration regarding to the heat treatment has been deleted.
L51 - improved stategy
- Modified (L82)
L54 - if it has been done in cheese before what is novel here?
- The enrichment with GPP was performed only for bovine dairy products obtained after fermentation with commercial starter cultures. In our work we used selected starter cultures and ewes’ milk. Furthermore, our paper reports novel data regarding the sensory evaluation, the volatile organic compounds and the gastrointestinal digestion.
L66 removed second [.]
- Deleted (L97).
L76 why were you adding polyphenols is this separate to the prepared GPP?
- These sentence has been split and rewritten for higher clarity (L411-415). Change highlighted in yellow because first requested by reviewer 1. The addition of polyphenols to milk was performed in a previous work (Barbaccia et al., 2020) in order to select indigenous milk lactic acid bacteria grown in presence of the main grape polyphenols and after that screening the four Lc. lactis used in the present work were selected.
L124 - why was it necessary to genotype the different strains?
- The isolation, typing and identification was performed only for the lactic acid bacteria isolated able to survive at the pasteurization process. These analysis was performed in order to exclude the interference of indigenous LAB over the four selected Lactococcus lactis strains (Mise36, Mise94, Mise169 and Mise190).
L180 - were these trained assessors?
- This information has been added in the text (L248-250). Change highlighted in yellow because first requested by reviewer 1.
L220 - do you have a reference for this method?
- Added (line 301)
Section 3.2 can you provide greater clarity as to what this has to do with GPP inclusion or is there any link? This is not routine analysis?
- This section is independent and although this is not routine analysis was performed in order to evaluate the persistence of four selected LAB over the indigenous milk LAB resistant to pasteurization process. As explained above, it was performed to evaluate and monitor starter strain dominance and persistence.
Other than the type of cheese chosen what is the big difference between this study and that by Marchiani 2016?
- Thanks for your comment. In the work of Marchiani et al. (2016) bovine cheese have been obtained after fermentation with commercial starter cultures. This technology, from a mere microbiological perspective, can compromise the individual characteristics of the final product. Moreover, Marchiani et al. evaluate only the total phenolic content and radical scavenging activity of final cheeses. In our work, ovine cheeses were produced with selected indigenous milk lactic acid bacteria grown in presence of the main grape polyphenols. The addition of autochthonous starter cultures ensure the stability of the final cheese. Furthermore, our paper reports novel data regarding the sensory evaluation, the volatile organic compounds and the gastrointestinal digestion.
Section 3.7 - grammar in this section needs to be reviewed.
- Following the new numeration this section is now section 3.8.The section has been reviewed.
Is this in vitro model of membrane lipid oxidation typically used?
- In our experience, brain microsomes in vitro peroxidation is a good model to assess antioxidant properties of compounds or food extracts (see ref. 29).
L419 - very more? Significantly?
- Significantly (L598)
L431 - are there any comparison with existing literature how this compares with other foodstuffs or polyphenol enriched foods?
- Comments added in the text (L600-602)and a new reference has been provided (L795-796).
L436 - i think ripening would be a more accurate terminology.
- Changed (L605).
Round 2
Reviewer 1 Report
The manuscript has been improved and the authors replied in all comments.
Please consider the following minor comments:
Lines 440-441. “GPP are poor in lipid components, thus its inclusion in cheese decreased fat level and that of proteins”. Please provide the range of these concentrations and the appropriate reference.
Section 2.1. Please provide more information regarding grape pomace. It is very important to know if the grape pomace was taken before or after the completion of fermentation since this influences its composition.
Line 329. Correct to figure 3.
Lines 81-83. “To this purpose, the fortification of cheeses with non-dairy ingredients represents an improved strategy to enhance the functional and bioactive properties of the final products [9].” Please check carefully the references. For example [9] is not for non-dairy ingredients.
Please also consider, if necessary to use the following reference for the composition of grape pomace and winemaking by products in general (https://doi.org/10.3390/ASEC2020-07521) (https://doi.org/10.3390/foods9111627)
Author Response
The manuscript has been improved and the authors replied in all comments.
- Thanks for your comments. All your suggestions were considered and the changes in the text were highlighted in yellow.
Please consider the following minor comments:
Lines 440-441. “GPP are poor in lipid components, thus its inclusion in cheese decreased fat level and that of proteins”. Please provide the range of these concentrations and the appropriate reference.
- We apologize for the mistake. The sentence has been modified as follow: “GPP are poor in lipid components; thus, GPP inclusion in cheese decreased fat level and, as a consequence, protein content increased”. The levels of fat and protein in both control and experimental cheese have been added in the text (L339-344).
Section 2.1. Please provide more information regarding grape pomace. It is very important to know if the grape pomace was taken before or after the completion of fermentation since this influences its composition.
- We agree with your comment and the sentence was modified to address your request (L72-73).
Line 329. Correct to figure 3.
- Corrected (L267)
Lines 81-83. “To this purpose, the fortification of cheeses with non-dairy ingredients represents an improved strategy to enhance the functional and bioactive properties of the final products [9].” Please check carefully the references. For example [9] is not for non-dairy ingredients.
- We apologize for the mistake. The reference has been changed in the text (L53) and in the references list (L530-531).
Please also consider, if necessary to use the following reference for the composition of grape pomace and winemaking by products in general (https://doi.org/10.3390/ASEC2020-07521) (https://doi.org/10.3390/foods9111627)
- The missing citations have been added in the text (L40-41) and in the references list (L517-518, 519-520).

Reviewer 3 Report
Most of my comments have now been addressed.
I would add that based on the article and comments recieved back that the sections on strains characterisation etc dont have much to do with the hypothesise of the effect of GPP on final product but rather impact of processing on their growth.. However if the other reviewers, possibly more well versed in this area than I, dont have an issue with this then article should proceed.
Author Response
Most of my comments have now been addressed.
I would add that based on the article and comments recieved back that the sections on strains characterisation etc dont have much to do with the hypothesise of the effect of GPP on final product but rather impact of processing on their growth.. However if the other reviewers, possibly more well versed in this area than I, dont have an issue with this then article should proceed.
- Probably the text was not clear enough. Strain typing was performed to distinguish the indigenous LAB resistant to the pasteurization from the starter strains inoculated to carry out the acidification process. This part is important, since without the polymorphic profiles generated by the RAPD-PCR analysis, the direct comparison of the dominant strains could not be possible. Please consider (L146-150)
